# CARD8 inflammasome activation during HIV-1 cell-to-cell transmission

Jessie Kulsuptrakul[1,2], Michael Emerman[2]*, Patrick S Mitchell[3,4]*

[1]Molecular and Cellular Biology Graduate Program, University of Washington, Seattle, United States; [2]Divisions of Human Biology and Basic Sciences, Fred Hutchinson Cancer Center, Seattle, United States; [3]Department of Microbiology, University of Washington, Seattle, United States; [4]Howard Hughes Medical Institute, University of Washington, Seattle, United States

## eLife Assessment

In this study, the authors investigated whether HIV-1 cell-to-cell transmission activates the CARD8 inflammasome in macrophages. The data **convincingly** support the idea that CARD8 is activated by the viral protease, promoting inflammation. The study's significance is further enhanced by including time-course analyses in primary T cells and macrophages and provides **valuable** insights into the role of CARD8 in HIV-induced inflammation.

*For correspondence:
memerman@fredhutch.org (ME);
psmitche@uw.edu (PSM)

Competing interest: The authors declare that no competing interests exist.

**Abstract** Our previous work demonstrated that CARD8 detects HIV-1 infection by sensing the enzymatic activity of the HIV protease, resulting in CARD8-dependent inflammasome activation (Kulsuptrakul et al., 2023). CARD8 harbors a motif in its N-terminus that functions as a HIV protease substrate mimic, permitting innate immune recognition of HIV-1 protease activity, which when cleaved by HIV protease triggers CARD8 inflammasome activation. Here, we sought to understand CARD8 responses in the context of HIV-1 cell-to-cell transmission via a viral synapse. We observed that cell-to-cell transmission of HIV-1 between infected T cells and primary human monocyte-derived macrophages induces CARD8 inflammasome activation in a manner that is dependent on viral protease activity and largely independent of the NLRP3 inflammasome. Additionally, to further evaluate the viral determinants of CARD8 sensing, we tested a panel of HIV protease inhibitor-resistant clones to establish how variation in HIV protease affects CARD8 activation. We identified mutant HIV-1 proteases that differentially cleave and activate CARD8 compared to wildtype HIV-1, thus indicating that natural variation in HIV protease affects not only the cleavage of the viral Gag-Pol polyprotein but also likely impacts innate sensing and inflammation.

## Introduction

HIV-1 disease progression is characterized by chronic inflammation, immune activation, CD4+ T cell depletion and eventual destruction of the immune system and susceptibility to opportunistic infections. The primary cellular targets of HIV-1 are activated CD4+ T helper cells, specialized CD4+ T cell subtypes such as Th17 cells (*Brenchley et al., 2008*; *Gosselin et al., 2010*; *Rodriguez-Garcia et al., 2014*), central memory cells (*Chun et al., 1997*; *Chun et al., 1995*), and macrophages (*Collman et al., 1990*; *Collman et al., 1989*). Chronic immune activation is primarily caused by rapid depletion of mucosal Th17 cells responsible for maintaining gut epithelial barrier integrity (*Brenchley et al., 2008*; *Brenchley et al., 2006*). In addition to inflammation induced by circulating microbial ligands, inflammation can also originate from HIV-infected cells through activation of innate immune sensors that form cytosolic immune complexes known as inflammasomes. Inflammasome activation ultimately results in

activation of pro-inflammatory caspases including caspase 1 (CASP1). Active CASP1 processes inflammatory cytokines and activates the pore-forming protein gasdermin D (GSDMD), which forms small pores in the plasma membrane and initiates a lytic form of cell death known as pyroptosis and the release of mature inflammatory cytokines interleukin (IL)-1β and IL-18 (*Broz and Dixit, 2016*; *Fink and Cookson, 2005*).

In prior work, we and others showed that the inflammasome-forming sensor CARD8 senses HIV-1 infection through the detection of HIV-1 protease (HIV^PR) activity (*Clark et al., 2023*; *Kulsuptrakul et al., 2023*; *Wang et al., 2021*). While the canonical function of HIV^PR is to cleave viral polyproteins during virion maturation, active HIV^PR is also released into the host cell, which is sensed by CARD8 via HIV^PR cleavage of its N-terminus and subsequent inflammasome activation. In this way, the CARD8 N-terminus functions as a 'molecular tripwire' to recognize the enzymatic activity of HIV^PR and other viral proteases (*Castro and Daugherty, 2023*; *Nadkarni et al., 2022*; *Tsu et al., 2023*). Moreover, HIV^PR cleavage of CARD8 occurs rapidly after infection such that HIV^PR inhibitors and fusion inhibitors, but not reverse transcriptase (RT) inhibitors can prevent CARD8 inflammasome activation, implying that CARD8 detects HIV-1 viral protease activity of virion-packaged or 'incoming' HIV^PR upon virion fusion (*Kulsuptrakul et al., 2023*; *Wang et al., 2024*; *Wang et al., 2021*). Interestingly, CARD8 inflammasome activation in resting CD4+ T cells results in pyroptosis but not the release of pro-inflammatory cytokines IL-1β or IL-18 (*Wang et al., 2024*), suggesting that CARD8 inflammasome activation in T cells does not directly contribute to chronic inflammation. Here, we address whether or not CARD8 may influence HIV-1 pathogenesis through the maturation and release of IL-1β from infected macrophages.

HIV-1 can be transmitted from one cell to another via two main mechanisms: 'cell-free' infection through binding of free HIV-1 virions to target cells, and cell-to-cell infection whereby infected cells directly transfer virus to an uninfected target cell via the formation of a transient viral synapse (*Chen et al., 2007*; *Galloway et al., 2015*; *Iwami et al., 2015*). Cell-to-cell transmission of HIV-1 has been reported between multiple HIV-1 target cell types including between active and resting CD4+ T cells (*Agosto et al., 2018*; *Martin et al., 2010*) and between CD4+ T cells and macrophages (*Baxter et al., 2014*; *Dupont and Sattentau, 2020*; *Lopez et al., 2019*). Cell-to-cell transmission delivers a large influx of virus to target cells, resulting in a high multiplicity of infection (MOI) (*Agosto et al., 2015*; *Del Portillo et al., 2011*; *Duncan et al., 2013*; *Russell et al., 2013*), which has been proposed to enhance viral fitness by overwhelming host restriction factors including Tetherin/BST-2 (*Jolly et al., 2010*; *Zhong et al., 2013*), SAMHD1 (*Xie et al., 2019*), and TRIM5α (*Richardson et al., 2008*), and evading adaptive immune responses including broadly neutralizing antibodies (*Abela et al., 2012*; *Dufloo et al., 2018*). Cell-to-cell spread of HIV-1 is thus an important consideration in studying CARD8 inflammasome activation.

Here, we investigated both host and viral determinants of CARD8 inflammasome activation upon HIV-1 infection. We evaluated CARD8 sensing of HIV^PR during cell-to-cell transmission of HIV-1 from T cell lines to myeloid cells in both immortalized and primary cell models of infection. We found that CARD8 inflammasome activation occurs in the context of cell-to-cell transmission from both SUPT1 cells, a T cell lymphoma cell line, to THP-1 cells, an acute myeloid leukemia cell line, and from primary CD4+ T cells to primary monocyte-derived macrophages. We also observed that HIV-triggered CARD8 inflammasome activation is largely independent of the NLRP3 inflammasome, which has previously been implicated in innate sensing of HIV-1 (*Bandera et al., 2018*; *Chivero et al., 2017*; *Hernandez et al., 2014*; *Leal et al., 2020*; *Mamik et al., 2017*; *Zhang et al., 2021*). Our findings suggest that CARD8 sensing of HIV^PR activity during cell-to-cell transmission of HIV-1 to macrophages, leading to robust secretion of IL-1β, may be a source of inflammatory cytokines that promote pathogenic chronic inflammation and disease progression. In addition, we also show that natural variation in HIV^PR due to resistance to protease inhibitors also affects CARD8 cleavage and subsequent inflammasome activation. Our results extend the role of incoming HIV^PR on CARD8-dependent inflammasome activation of inflammasome responses as a function of cell type, mode of transmission, and virus evolution in response to antiviral therapy.

## Results

### Cell-to-cell transmission of HIV-1 induces CARD8 inflammasome activation

Our previous work investigating HIV-dependent CARD8 inflammasome activation used the cationic polymer DEAE-dextran, which is a common reagent used to enhance viral infection in cell culture (*Bailey et al., 1984*). However, we found that DEAE-dextran could induce inflammasome activation in the absence of viral infection in some 'wildtype' (WT) THP-1 cell stocks (see Appendix 1). These results prompted us to establish other models of HIV-1 infection and subsequent inflammasome activation that lack cationic polymers. Thus, we designed an *in vitro* coculture infection system to mimic HIV-1 cell-to-cell transmission by infecting SUPT1 cells, a T-cell lymphoma line (i.e., donor cells) and

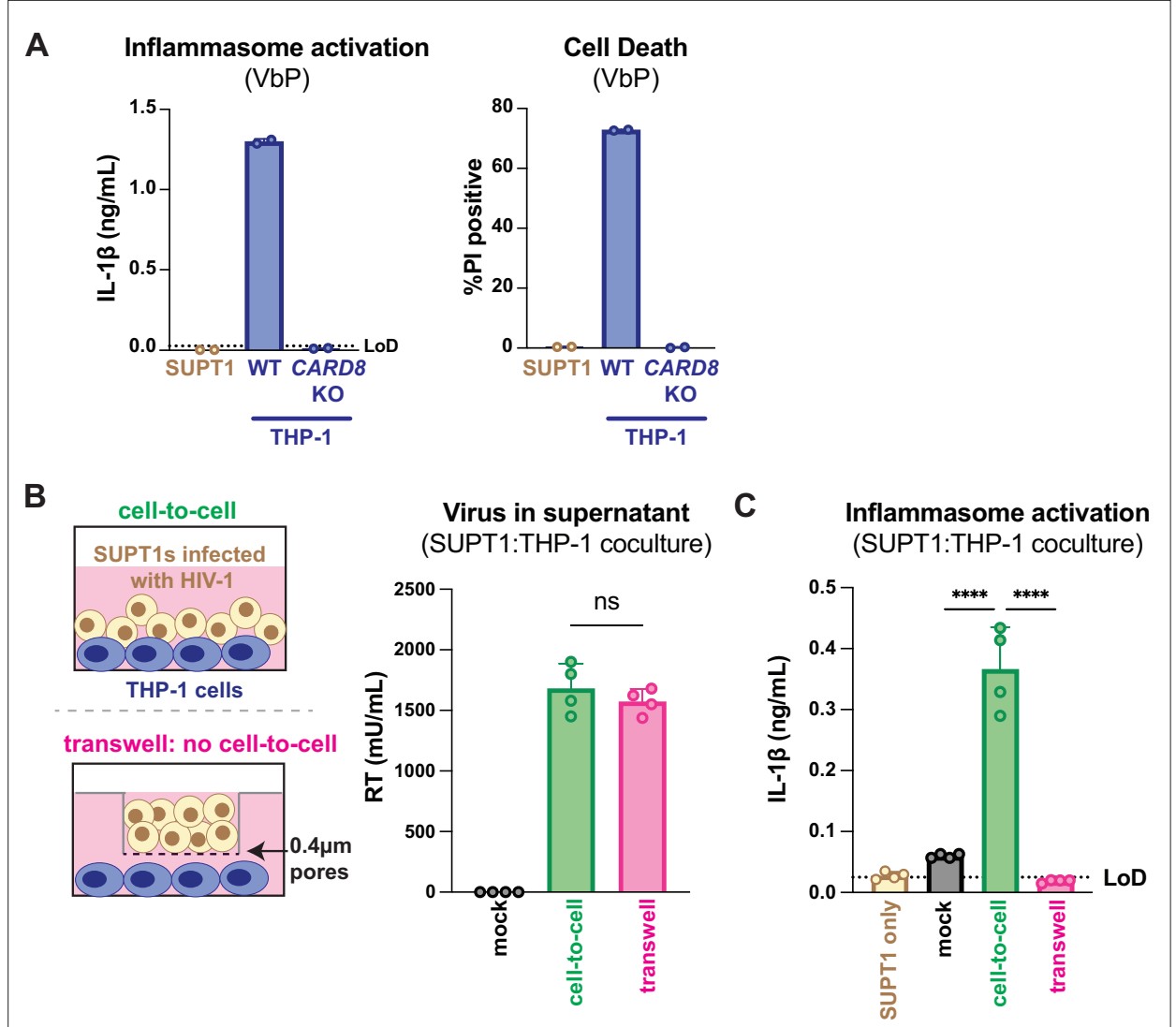

**Figure 1.** HIV-1 cell-to-cell infection induces inflammasome activation. (**A**) SUPT1 or THP-1 cells were primed with Pam3CSK4 (500ng/mL) overnight then treated with 5 µM ValboroPro (VbP) for 24 hours, then assessed for IL-1β secretion and cell death via propidium iodide (PI) uptake. %PI positive was normalized to mock-infected controls. (**B**) (left) Schematic illustrating the experimental setup for SUPT1:THP-1 cell coculture either with (bottom) or without (top) a transwell. (right) SUPT1 cells were either mock-infected or infected with HIV-1$_{LAI}$ then cocultured with primed WT THP-1 cells (see Methods) 20 hours post infection. Mock- or HIV-1$_{LAI}$-infected SUPT1 cells were either mixed with the THP-1 cells or put in a transwell with a virus-permeable membrane as shown in panel (**B**) (left). Supernatant in the cell-to-cell condition and in the supernatant outside of the transwell were sampled and measured for released HIV virions via a reverse transcriptase (RT) assay or (**C**) IL-1β secretion 3 days after starting the coculture. Dotted line indicates limit of detection (LoD). Datasets represent mean ± SD (**A**: n=2; **B, C** n=4 biological replicates). One-way ANOVA with (**B**) Tukey's or (**C**) Dunnett's test using GraphPad Prism 10. ns = not significant, *p<0.05, **p<0.01, ***p<0.001, ****p<0.0001.

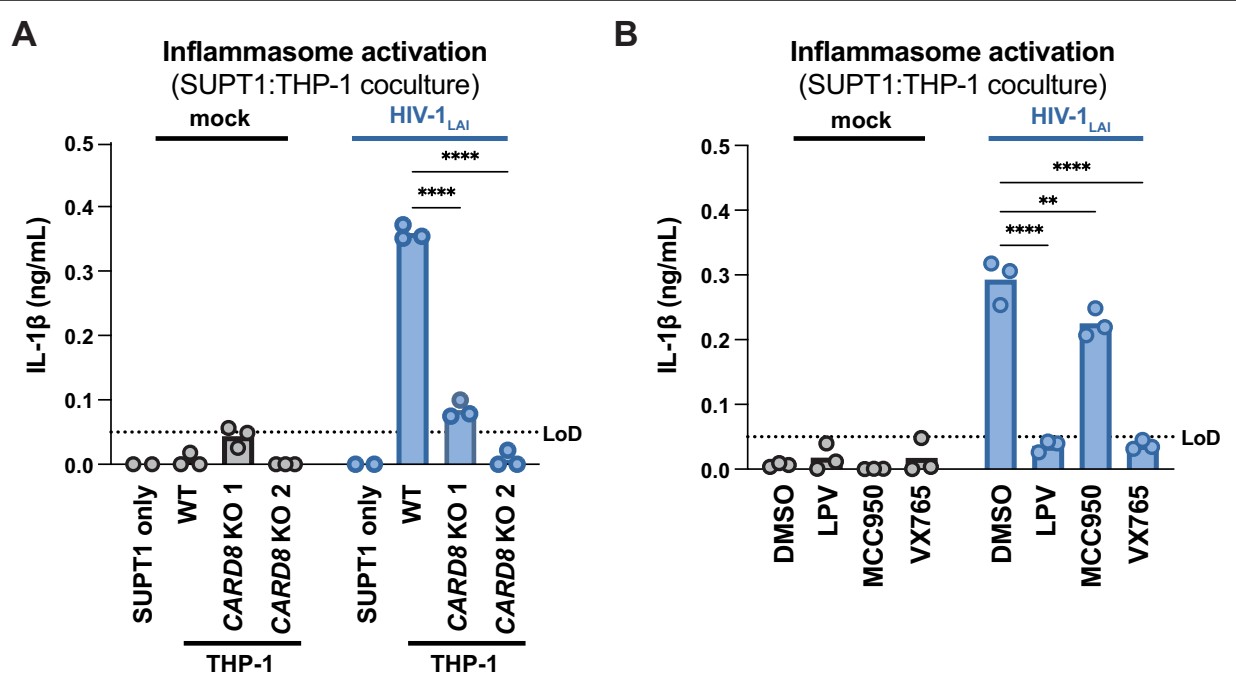

**Figure 2.** HIV-1 cell-to-cell transmission induces CARD8-dependent activation largely independent of NLRP3. (**A**) SUPT1 cells were either mock-infected or infected with HIV-1$_{LAI}$ for 18–20 hours prior to coculture with wildtype (WT) or *CARD8* knockout (KO) THP-1 cells. The coculture was harvested 72 hours later to probe for IL-1β secretion in the coculture supernatant via IL-1R reporter assay. THP-1 cells were primed with Pam3CSK4 (500ng/mL) for 16–24 hours prior to coculture. SUPT1 cells were infected with HIV-1$_{LAI}$ such that 30% of the cells were positive for intracellular p24$^{gag}$ after 18–20 hours. (**B**) SUPT1 cells were either mock- or HIV-1$_{LAI}$-infected as in (**A**) for 18–20 hours then incubated in DMSO, lopinavir (LPV), MCC950, or VX765 at 0.01%, 5 μM, 10μM, or 1μg/mL, respectively, for 15 minutes prior to coculturing with primed WT THP-1 cells. The coculture was assessed for subsequent inflammasome activation after 72 hours as in (**A**). Dotted line indicates limit of detection (LoD). Datasets represent mean ± SD (n=3 biological replicates). Two-way ANOVA with Dunnett's test using GraphPad Prism 10. ns = not significant, *p<0.05, **p<0.01, ***p<0.001, ****p<0.0001.

The online version of this article includes the following figure supplement(s) for figure 2:

**Figure supplement 1.** HIV-dependent inflammasome activation is largely NLRP3-independent.

then mixing them with uninfected THP-1 cells (i.e., target cells). We opted for SUPT1 cells as the viral producer cell line because they are permissive to HIV-1 infection, and unlike THP-1 cells, SUPT1 cells do not respond to a known CARD8 inflammasome activator, ValboroPro (VbP), as assayed by both IL-1β secretion and cell death, indicating that SUPT1 cells do not have a functional CARD8 inflammasome pathway (*Figure 1A*). This allowed us to infer that inflammasome outputs (e.g., IL-1β secretion) in our coculture system occur upon cell-to-cell transmission of HIV-1 from SUPT1 cells to the CARD8-competent THP-1 cells.

We found that coculture of THP-1 cells with HIV-1$_{LAI}$-infected SUPT1 cells (*Figure 1B*) but not mock-infected SUPT1 cells results in robust inflammasome activation as indicated by IL-1β secretion, suggesting that our coculture system, which lacks DEAE-dextran, can induce HIV-dependent inflammasome activation via cell-to-cell infection (*Figure 1C*). To further test this assumption, we prevented cell-to-cell contact using a virus-permeable transwell with a 0.4 μm pore insert (*Figure 1B*, left). We verified that there were equivalent amounts of infectious virus in the cell-to-cell condition versus the lower chamber of the transwell condition by measuring RT activity in the supernatant (*Figure 1B*, right). Despite equivalent amounts of infectious virus in both conditions, we observed that HIV-1$_{LAI}$-infected SUPT1 cells (upper chamber) cocultured with THP-1 cells (lower chamber) did not lead to detectable IL-1β secretion (*Figure 1C*). These data suggest that our SUPT1:THP-1 coculture system can trigger HIV-dependent inflammasome activation in a manner dependent on cell-to-cell contact.

We next assessed the role of CARD8 and other inflammasome sensors during cell-to-cell transmission of HIV-1. We cocultured mock- or HIV-1$_{LAI}$-infected SUPT1 cells with either WT or *CARD8* KO THP-1 cells and compared inflammasome activation by measuring levels of secreted IL-1β. HIV-1$_{LAI}$-infected SUPT1 cells cocultured with WT but not *CARD8* KO THP-1 cells resulted in a significant

increase in IL-1β (*Figure 2A*). These results suggest that CARD8 is the primary sensor that drives inflammasome activation in HIV-1 cell-to-cell transmission to THP-1 cells. Since the NLRP3 inflammasome has previously been implicated in HIV-dependent inflammasome activation (*Bandera et al., 2018*; *Chivero et al., 2017*; *Hernandez et al., 2014*; *Leal et al., 2020*; *Mamik et al., 2017*; *Zhang et al., 2021*), we also assessed the effects of the NLRP3 inflammasome-specific inhibitor MCC950 (*Coll et al., 2015*; *Primiano et al., 2016*) on inflammasome activation in our coculture system. Treatment with MCC950 or the caspase 1 (CASP1) inhibitor VX765 (*Wannamaker et al., 2007*) were sufficient to abrogate inflammasome activation induced by the ionophore nigericin, a well-characterized NLRP3 agonist (*Figure 2—figure supplement 1A*). However, in the HIV-1 coculture system, MCC950 treatment had only a modest effect on inflammasome activation while VX765 and the HIV$^{PR}$ inhibitor lopinavir (LPV), which prevents CARD8 cleavage by HIV-1$^{PR}$ (*Kulsuptrakul et al., 2023*; *Wang et al., 2021*), completely abrogated IL-1β secretion (*Figure 2B*). We observed similar results during HIV-1$_{LAI}$ and HIV-1$_{LAI-VSVG}$ cell-free infection of THP-1 cells in the presence of DEAE-dextran (*Figure 2—figure supplement 1B*). Taken together, these findings indicate that HIV-dependent inflammasome activation via cell-to-cell transmission is CARD8-dependent and largely NLRP3-independent.

## CARD8, but not NLRP3, is required for inflammasome activation during HIV-1 cell-to-cell transmission into primary monocyte-derived macrophages

We next examined inflammasome activation upon HIV-1 cell-to-cell transmission in primary human monocyte-derived macrophages (MDMs). Previously, we had observed that CARD8 could sense active HIV-1$^{PR}$ released into the host cytosol following viral fusion, which we refer to as 'incoming' HIV-1$^{PR}$ in our cell-free infection system in THP-1 cells using DEAE-dextran and spinoculation (*Kulsuptrakul et al., 2023*). Thus, we assessed the importance of viral entry by coculturing MDMs from three independent donors with mock-, HIV-1$_{LAI}$-, or HIV-1$_{NL4.3-BaL}$-infected SUPT1 cells expressing CCR5 (SUPT1-CCR5) and assayed for inflammasome activation (*Figure 3A*). HIV-1$_{LAI}$ is a CXCR4 tropic strain unable to infect macrophages, whereas HIV-1$_{NL4.3-BaL}$ uses CCR5 as a co-receptor which is required for infection of macrophages. We observed inflammasome activation, as measured by IL-1β secretion, in MDMs cocultured with HIV-1$_{NL4.3-BaL}$-infected SUPT1 cells but not in MDMs cocultured with mock- or HIV-1$_{LAI}$-infected SUPT1-CCR5 cells (*Figure 3A*). This demonstrates that HIV-dependent inflammasome activation can occur in MDMs during cell-to-cell infection in a manner dependent on viral entry. To further ascertain if this inflammasome activation was CARD8-dependent and driven by incoming HIV$^{PR}$ during SUPT1:MDM cell-to-cell transmission, we investigated the effects of different inhibitors on inflammasome activation in MDM cocultures with HIV-1$_{NL4.3-BaL}$-infected SUPT1s. We observed that IL-1β secretion was abrogated by treatment with lopinavir, an HIV-1 protease inhibitor, and VX765, a CASP1 inhibitor, indicating that inflammasome activation in MDM cocultures is dependent on HIV$^{PR}$ and CASP1, respectively (*Figure 3A*). In addition, we used an RT inhibitor, nevirapine (NVP), to prevent synthesis of *de novo* translated HIV$^{PR}$, and thus any CARD8-dependent IL-1β secretion would only be due to incoming HIV$^{PR}$ in the presence of NVP. Indeed, we observed HIV-dependent inflammasome activation in the presence of NVP that was added at the time of coculture, indicating that incoming HIV$^{PR}$ is sufficient to elicit an inflammasome response (*Figure 3A*). Lastly, MDM cocultures treated with MCC950, an inhibitor of the NLRP3 inflammasome, had no effect on IL-1β secretion (*Figure 3A*), and we observed similar inflammasome activation results regardless of whether the MDMs were primed with TLR1/2 agonist Pam3CSK4 versus TLR4 agonist lipopolysaccharide (LPS) (*Figure 3—figure supplement 1A*). Thus, cell-to-cell contact of infected cells with primary monocyte-derived macrophages can elicit an inflammasome response in a manner that is dependent on viral entry, CASP1, and incoming HIV$^{PR}$, and independent from NLRP3.

To further assess the timing of this inflammasome activation, we conducted a SUPT1:MDM time-course coculture experiment with three additional donors, assaying IL-1β secretion at 4, 24, 48, and 72 hours post coculture in the presence or absence of NVP. We observed inflammasome activation as determined by IL-1β secretion by 24 hours post coculture in all donors that persisted at a similar level at 48 and 72 hours post coculture (*Figure 3B*). As expected, we observed donor-to-donor variation in the extent to which IL-1β secretion occurred following HIV-1 infection. However, per donor, HIV-1-driven IL-1β levels were comparable to VbP-induced inflammasome activation (*Figure 3—figure supplement 1B*). Moreover, adding NVP had no effect on IL-1β secretion in HIV-1-infected

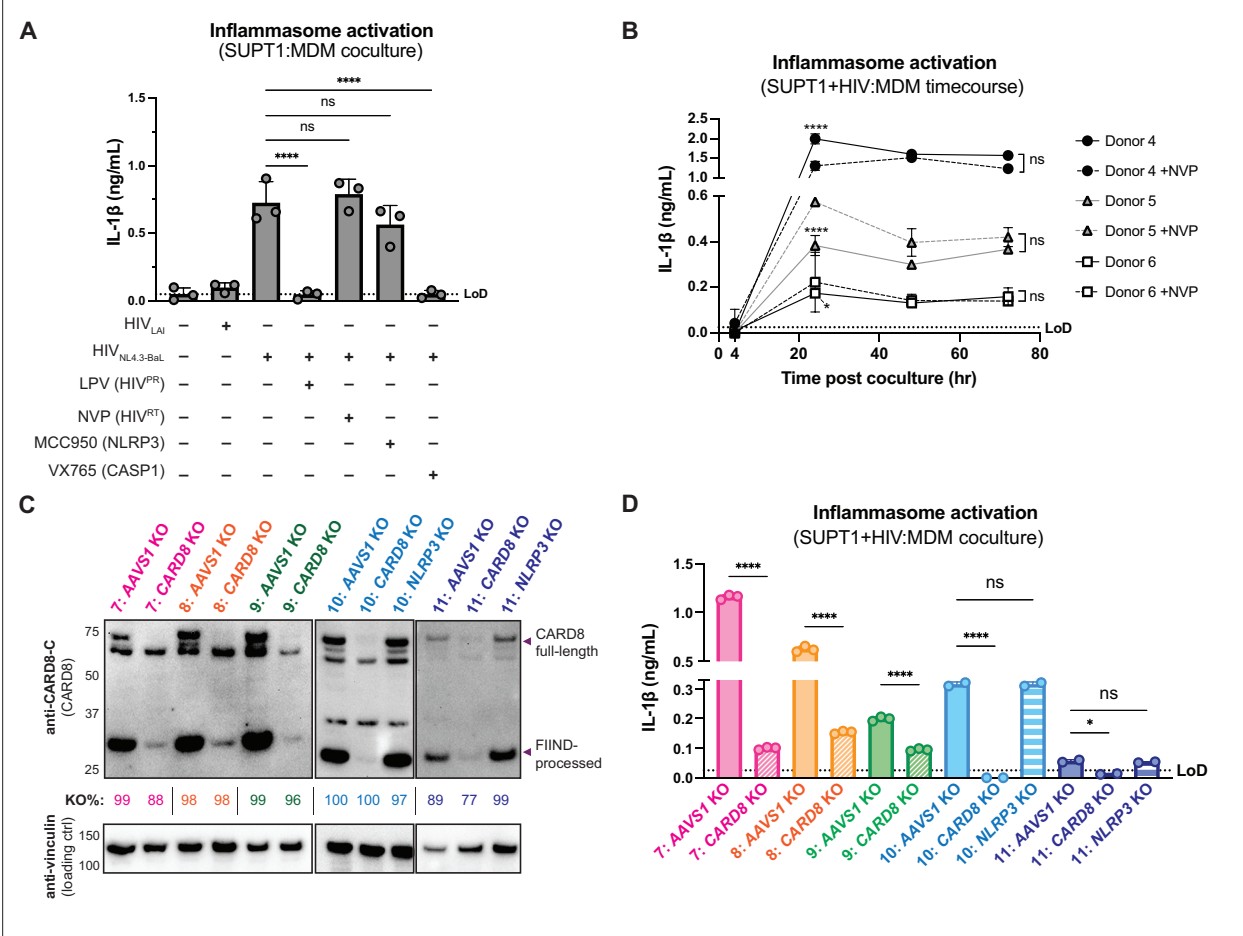

**Figure 3.** Cell-to-cell HIV infection induces CARD8-dependent inflammasome activation in monocyte-derived macrophages (MDMs). (**A**) MDMs from three independent donors were cocultured with SUPT1 cells expressing CCR5 (SUPT1-CCR5) that were mock-, HIV-1$_{LAI}$-, or HIV-1$_{NL4.3-BaL}$-infected then assayed for inflammasome activation 48 hours post coculture for IL-1β secretion. Fifteen minutes before starting the coculture, SUPT1-CCR5 cells infected with HIV-1$_{NL4.3-BaL}$ were pre-treated with either DMSO, lopinavir (5μM), nevirapine (50 μM), MCC950 (10 μM), or VX765 (1μg/mL), inhibiting HIV-1 protease (HIV$^{PR}$), HIV-1 reverse transcriptase (HIV$^{RT}$), NLRP3, or caspase 1 (CASP1), respectively. (**B**) MDMs from three independent donors were cocultured with SUPT1-CCR5 cells infected with HIV-1$_{NL4.3-BaL}$ in either the presence or absence of nevirapine (NVP). Supernatant was harvested at 4, 24, 48, or 72 hours to assay for IL-1β secretion. (**C**) MDMs from five independent donors were knocked out (KO) for *AAVS1 CARD8,* or *NLRP3* using a Synthego gene KO kit then immunoblotted using an anti-CARD8 antibody or anti-vinculin. Full-length and FIIND-processed CARD8 intermediates are marked with a purple arrow. Size is indicated in kDa on left side of blot. Table between CARD8 and vinculin blot shows Synthego gene KO% scores for each donor KO line. (**D**) *AAVS1*, *CARD8* or *NLRP3* KO MDM lines from (**C**) were primed with Pam3CSK4 (500 ng/mL) overnight and then cocultured with SUPT1-CCR5 cells mock-, or HIV-1$_{NL4.3-BaL}$-infected then assayed for inflammasome activation 48 hours post coculture for IL-1β secretion. For all SUPT1:MDM experiments, SUPT1-CCR5 cells were infected with HIV-1$_{LAI}$ or HIV-1$_{NL4.3-BaL}$ such that 5–20% of cells were positive for intracellular p24$^{gag}$ after 20 hours. IL-1 levels shown were normalized to the SUPT1 mock-infected coculture control. Dotted line indicates limit of detection (LoD). Datasets represent mean ± SD (**A**: n=3 independent donors, **B**: n=2 biological replicates for each donor, **D**: n=3 technical replicates per donor). One-way ANOVA with (**A**) Tukey's or (**D**) Sidak's test or (**B**) two-way ANOVA with Tukey's test using GraphPad Prism 10. ns = not significant, *p<0.05, **p<0.01, ***p<0.001, ****p<0.0001.

The online version of this article includes the following source data and figure supplement(s) for figure 3:

**Source data 1.** Original western blot images used to generate *Figure 3C*.

**Source data 2.** Original western blot .tif files used to generate *Figure 3C*.

**Figure supplement 1.** Cell-to-cell HIV infection induces CARD8-dependent activation in monocyte-derived macrophages (MDMs).

MDM cocultures (*Figure 3B*). To verify that NVP was functional, we assayed the supernatant of the mock- and NVP-treated cocultures from *Figure 3B* at 48 hours post coculture for infectious virions via an assay that measures RT activity and observed lower RT activity in NVP-treated MDM donors (*Figure 3—figure supplement 1C*). Unlike the 'cell-free' infection conditions in which we previously

observed an increase in IL-1β levels 4 hours post-infection, we did not detect measurable differences in IL-1β secretion at this early timepoint following the establishment of SUPT1:MDM coculture. Nevertheless, the data are consistent with incoming HIV$^{PR}$ being responsible for inflammasome activation during cell-to-cell transmission of HIV because the induction of IL-1β persists in the presence of NVP which would block any *de novo* synthesis of new *gag/pol* products (*Figure 3A and B*). Taken together, these data suggest that in the context of cell-to-cell transmission, CARD8 is likely the inflammasome-forming sensor that detects HIV-1 infection via incoming HIV$^{PR}$ activity in primary monocyte-derived macrophages.

To specifically address the role of CARD8 in HIV-1-induced inflammasome activation in MDMs, we genetically edited MDMs by isolating monocytes from five donors and electroporated them with Cas9 RNPs complexed with three unique sgRNAs per gene targeting *AAVS1*, a safe harbor locus, *CARD8* or *NLRP3* (only two donors for *NLRP3* KO). Edited MDMs were then differentiated for 6 days prior to evaluating KO efficiency and initiating cocultures with HIV-1-infected SUPT1 cells. To verify KO efficiency, we immunoblotted with an antibody that detects the CARD8 C-terminus in *CARD8* KO MDMs relative to the *AAVS1* KO control and observed a marked reduction of the full-length and FIIND-processed CARD8 protein in all five donors (*Figure 3C*). In addition, we confirmed *AAVS1*, *CARD8*, and *NLRP3* KO at the genetic level via Synthego ICE analysis (*Conant et al., 2022*), measuring >85% KO efficiency (*Figure 3C*). We also observed robust inflammasome activation upon treatment with CARD8 inflammasome activator VbP as measured by IL-1β secretion in *AAVS1* KO MDMs from two of the three donors, which was completely abrogated in *CARD8* KO MDMs, confirming functional loss of CARD8 (*Figure 3—figure supplement 1D*). We then cocultured either *AAVS1* KO, *CARD8* KO, or *NLRP3* KO MDMs with mock- or HIV-1$_{NL4.3-BaL}$-infected SUPT1-CCR5 cells at a 1:1 ratio and measured inflammasome activation via IL-1β secretion 48 hours post coculture. In all five donors, we observed significant reduction in inflammasome activation in *CARD8* KO cocultures relative to the *AAVS1* KO control (*Figure 3D*). Consistent with our findings with SUPT1:THP-1 coculture, we observed no difference in inflammasome activation when coculturing infected SUPT1 cells with *AAVS1* KO vs *NLRP3* KO MDMs, suggesting that the inflammasome activation was also largely NLRP3-independent (*Figure 3D*). Taken together, these data demonstrate that CARD8 is required for inflammasome activation in MDMs during HIV-1 cell-to cell transmission.

## Coculture of HIV-1-infected primary CD4+ T cells with primary MDMs elicits CARD8-dependent inflammasome activation

We next investigated inflammasome activation in the context of cell-to-cell infection using primary CD4+ T cells as donor cells, rather than SUPT1 cells, and primary MDMs as target cells. Mock- or HIV-1$_{NL4.3-BaL}$-infected CD4+ T cells were cocultured with MDMs in the presence or absence of various inhibitors (as in *Figure 3A*). We assessed inflammasome activation via IL-1β secretion 72 hours post coculture and observed inflammasome activation when coculturing with HIV-infected T cells but not mock-infected T cells or cocultures treated with LPV or VX765, demonstrating that inflammasome activation is driven by HIV$^{PR}$ and CASP1 (*Figure 4A*) and consistent with our findings using the SUPT1:MDM coculture (*Figure 3*). We also observed that inflammasome activation persisted in NVP-treated cocultures, suggesting that incoming protease is also important for inflammasome activation in the context of cell-to-cell transmission of HIV-1 from primary CD4+ T cells to primary MDMs (*Figure 4A*). To confirm the potency of LPV and NVP, we assayed for infectious virions in the supernatant of these CD4:MDM cocultures and detected a dramatic decrease in RT activity in the presence of either of these drugs, indicating that these drugs were efficacious at this dose in these cells (*Figure 4—figure supplement 1A*). We also conducted a time-course experiment with a CD4:MDM coculture from an independent donor in the presence or absence of LPV. Similar to the SUPT1:MDM time course (*Figure 3B*), we were able to detect elevated levels of IL-1β by 24 hours post coculture, which was again strictly dependent on the enzymatic activity of the viral protease as LPV treatment completely inhibited IL-1β secretion (*Figure 4B*).

To interrogate the specific role of CARD8 in this primary CD4:MDM coculture system, we generated *AAVS1* KO (as a control) or *CARD8* KO MDMs and cocultured the MDMs with either mock- or HIV-infected primary CD4+ T cells then assayed for inflammasome activation 48 hours post coculture. KO efficiency was confirmed via immunoblot and functional response to VbP (in one of the two donors) (*Figure 4C*, *Figure 4—figure supplement 1B*). We detected a significant decrease in

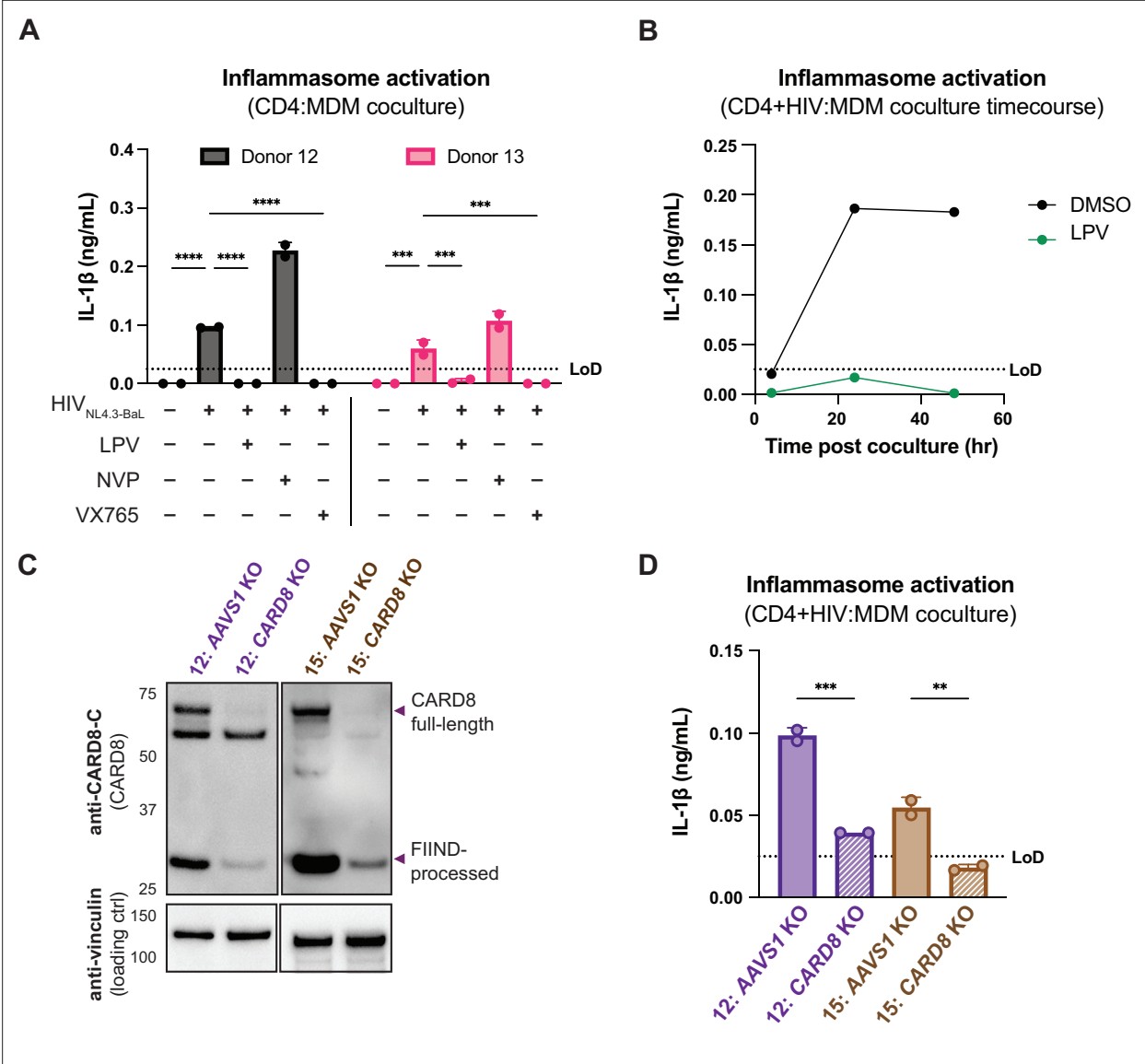

**Figure 4.** Primary CD4+ T cell:MDM coculture elicits CARD8-dependent inflammasome activation. (**A**) CD4+ T cells from a blood donor were isolated, activated, and either mock-infected or infected with HIV-1$_{NL4.3-BaL}$ for 3 days such that ~10% of cells were positive for intracellular p24$^{gag}$. Monocyte-derived macrophages (MDMs) were primed with Pam3CSK4 then cocultured with mock- or HIV-1-infected primary CD4 T cells in the presence or absence of lopinavir (LPV), nevirapine (NVP), or VX765, inhibiting HIV protease, reverse transcriptase, or caspase 1, respectively. Supernatants were harvested 3 days post coculture to assay for IL-1β secretion via IL-1 reporter assay. (**B**) CD4+ T cells from donor 12 and MDMs from donor 14 were cocultured as in (**A**) in the presence or absence of LPV. Supernatant was harvested at 4, 24 and 48 hours post coculture to probe for IL-1β secretion. (**C**) *AAVS1* or *CARD8* MDM KOs were immunoblotted using an anti-CARD8 antibody or anti-vinculin. Full-length and FIIND-processed CARD8 intermediates are marked with a purple arrow. Size is indicated in kDa on left side of blot. (**D**) *AAVS1* or *CARD8* KO MDMs from (**C**) were cocultured with CD4+ T cells infected with HIV-1$_{NL4.3-BaL}$ then assayed for IL-1β secretion 48 hours post coculture. The donor 12 cocultures consisted of autologous CD4s and MDMs, whereas the MDMs from donors 13–15 were cocultured with donor 12 CD4s. Dotted line indicates limit of detection (LoD). Datasets represent mean ± SD (**A, D**: n=2 technical replicates for each donor, **B**: n=3 technical replicates for one donor). (**A**) Two-way ANOVA with Tukey's test (**D**) One-way ANOVA with Sidak's test using GraphPad Prism 10. ns = not significant, *p<0.05, **p<0.01, ***p<0.001, ****p<0.0001.

The online version of this article includes the following source data and figure supplement(s) for figure 4:

**Source data 1.** Original western blot images used to generate *Figure 4C*.

**Source data 2.** Original western blot .tif files used to generate *Figure 4C*.

**Figure supplement 1.** Primary CD4 T cell:MDM coculture elicits CARD8-dependent inflammasome activation.

inflammasome activation when infected T cells were cocultured with *CARD8* KO versus *AAVS1* KO MDMs (*Figure 4D*), indicating that CARD8 is required for inflammasome activation in MDMs during HIV-1 transmission from primary CD4+ T cells to MDMs. Taken together, our data indicate that CARD8 plays a pivotal role in sensing and responding to HIV-1 cell-to-cell infection between primary CD4 T cells and macrophages.

## Protease inhibitor-resistant strains of HIV-1 differentially cleave and activate CARD8

The consequences of CARD8 inflammasome activation on viral replication have been challenging to assess given that viral fitness is intrinsically linked to viral protease processing of the viral polyprotein such that inhibiting HIV$^{PR}$ also prevents viral replication. In an attempt to circumvent this issue, we surveyed a panel of multi-HIV$^{PR}$ inhibitor-resistant (PI-R) infectious molecular clones of HIV-1 (*Varghese et al., 2013*). This panel of PI-R molecular clones varies in resistance to HIV protease inhibitors, including nelfinavir (NFV), fosamprenavir (FPV), saquinavir (SQV), indinavir (IDV), atazanavir (ATV), lopinavir (LPV), tipranavir (TPV), and darunavir (DRV). Each molecular clone encodes 4–11 mutations in HIV$^{PR}$ as well as various compensatory HIV$^{gag}$ mutations (*Varghese et al., 2013*; *Supplementary file 1*).

We initially tested if PI-R HIV-1 proviruses differentially cleave CARD8 by co-transfecting HEK293T cells with an expression plasmid encoding an N-terminal mCherry tagged human CARD8 and either wildtype HIV-1$_{LAI}$ or PI-R HIV-1 proviruses. HIV-1$_{LAI}$ protease cleaves CARD8 between phenylalanine (F) 59 and F60 (*Wang et al., 2021*), resulting in an ~33 kDa product (*Figure 5A*, top). By quantifying the 33 kDa CARD8 cleavage product with each HIV-1 provirus, we identified a PI-R clone that exhibited similar efficiency at cleaving CARD8 to HIV-1$_{LAI}$ (i.e., PI-R1), PI-R clones that were markedly less efficient at cleaving CARD8 than HIV-1$_{LAI}$ (i.e., PI-R2, PI-R3, PI-R5, PI-R9, and PI-R10) and two PI-R clones, PI-R12 and PI-R13, that were more efficient at cleaving CARD8 than HIV-1$_{LAI}$ (*Figure 5A*, top, *Supplementary file 1*). Of note, all PI-R proviruses had similar levels of HIV$^{PR}$ activity for HIV$^{gagpol}$ polyprotein processing from p55$^{gag}$ to p24$^{gag}$ as indicated by the ratio of p24$^{gag}$/p55$^{gag}$ quantified from the anti-p24$^{gag}$ immunoblot (*Figure 5A*, middle). These results indicate that naturally occurring HIV-1 protease mutations can influence host targets like CARD8.

We next assessed if PI-R clones exhibiting reduced (PI-R2 and -9) or increased (PI-R12 and -13) cleavage of CARD8 relative to HIV-1$_{LAI}$ (*Figure 5A and B*, *Supplementary file 1*) resulted in differential inflammasome activation. HEK293T cells endogenously express CARD8 but lack the downstream components (i.e., CASP1, GSDMD, and IL-1β/IL18) of the inflammasome pathway. Thus, we reconstituted the inflammasome pathway in HEK293T cells by co-transfection of human caspase 1, human pro-IL-1β, and either empty vector, HIV-1$_{LAI}$ or representative PI-R proviruses then quantified CASP1-dependent processing of pro-IL-1β as a readout of CARD8 inflammasome activation as in *Tsu et al., 2023*. Consistent with the observed differences in CARD8 cleavage by PI-R clones (*Figure 5A*), we found that PI-R2 and PI-R9, which exhibited less CARD8 cleavage than HIV-1$_{LAI}$, also induced lower IL-1β levels than HIV-1$_{LAI}$ (*Figure 5B*). Similarly, PI-R12 and PI-R13, which demonstrated enhanced CARD8 cleavage, elicited higher IL-1β levels than HIV-1$_{LAI}$ (*Figure 5B*). However, these PI-R clones, relative to the LAI strain, may have distinct protease substrate specificity, variable efficiency/kinetics in viral assembly, gag dimerization, and other factors, which may also influence CARD8 inflammasome activation. We next assessed inflammasome activation by the PI-R clones in a cell-to-cell transmission model using HEK293T cells as donor cells rather than SUPT1 cells and either WT or *CARD8* KO THP-1 cells as the target line at a 1:1 ratio. We opted to overexpress the HIV-1$_{LAI}$ or the PI-R proviruses in HEK293T cells rather than infecting SUPT1 cells due to dramatic variability in replication kinetics between PI-R strains. In these HEK293T:THP-1 cocultures, we observed that cell-to-cell transmission of PI-R2 and PI-R9 resulted in lower IL-1β levels while PI-R12 and PI-R13 resulted in higher IL-1β levels compared to HIV-1$_{LAI}$, respectively (*Figure 5C*), consistent with our findings from CARD8 cleavage (*Figure 5A*) and reconstituted inflammasome assays (*Figure 5B*). Our findings suggest that HIV-dependent inflammasome activation is under genetic control of the viral protease in a manner that can be increased or decreased with naturally occurring mutations induced by drug resistance.

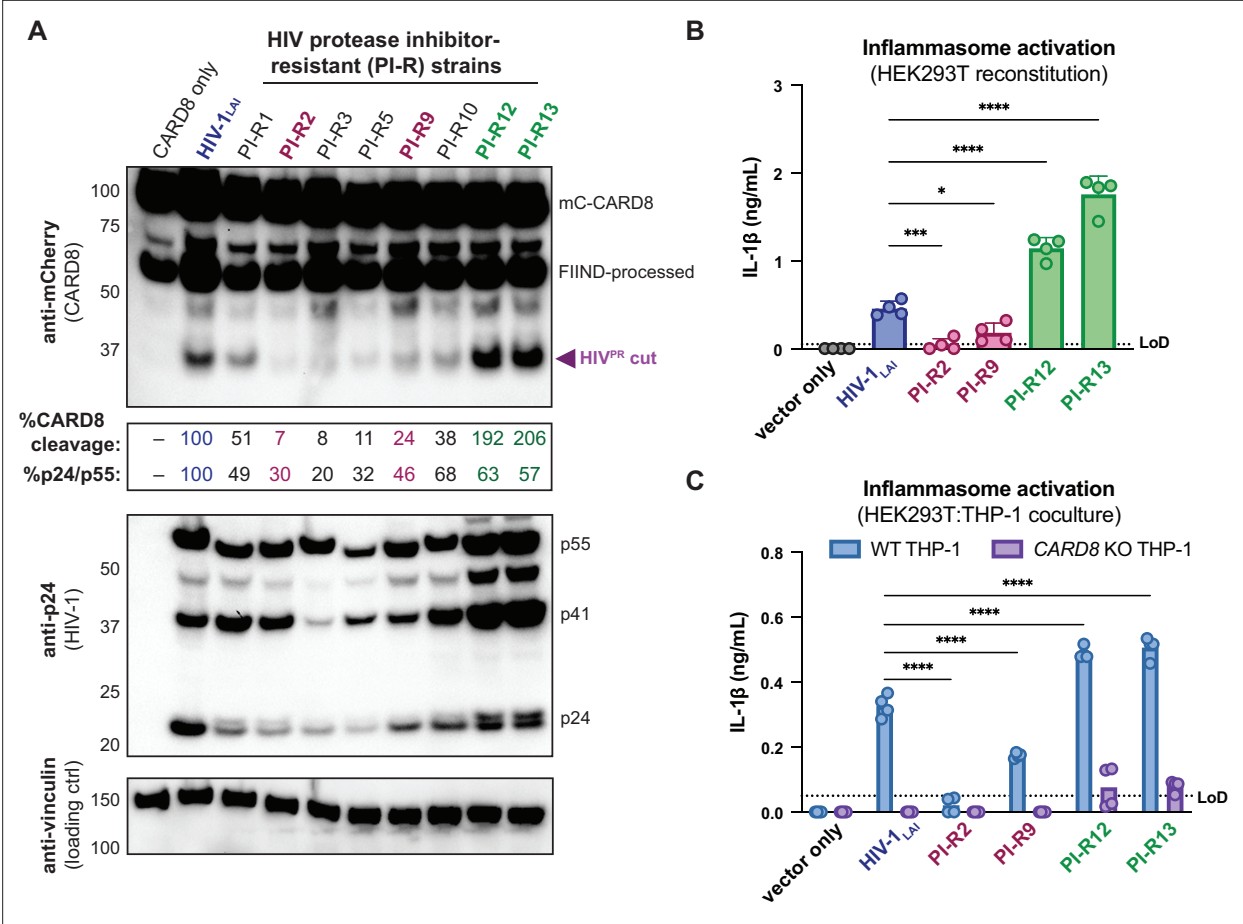

**Figure 5.** Protease inhibitor-resistant strains of HIV-1 differentially cleave and activate CARD8. (**A**) HEK293T cells were transfected with a construct encoding CARD8 with an N-terminal mCherry tag (mCherry-CARD8) and indicated HIV-1 proviral constructs. Protease inhibitor-resistant (PI-R) clones of HIV-1 are a subset of a panel expressing prototypical multidrug resistant HIV-1 protease (HIV[PR]) in an NL4.3 backbone (*Supplementary file 1*). Top: immunoblotting using anti-mCherry antibody to detect mCherry-CARD8. The full-length (mC-CARD8) and FIIND-processed bands are indicated as well as the HIV[PR] cut product. Size is indicated in kDa on left side of blot. The band at ~45 kDa is the result of cleavage by the 20S proteasome (*Hsiao et al., 2022*). % CARD8 cleavage was calculated by quantifying the HIV[PR] cut band relative to the HIV-1[LAI] control using BioRad Image Lab 6. Middle: immunoblotting with an anti-p24[gag] antibody showing HIV-1[gag] cleavage products p41[gag] and p24[gag], and/or full-length HIV-1[gag], p55[gag]. %p24/p55 was calculated from the ratio of p24[gag] versus p55[gag] product by quantifying the volume of the p24[gag] bands versus the p55[gag] band relative to the HIV-1[LAI] control using BioRad Image lab 6. Bottom: immunoblotting with an anti-vinculin antibody to detect vinculin as a loading control. (**B**) HEK293T cells were transfected with human caspase 1 and human pro-IL-1β, and either carrier vector or indicated HIV-1 proviruses then probed for IL-1β secretion 24 hours post-transfection via IL-1R reporter assay. (**C**) HEK293T cells were transfected with indicated HIV-1 proviruses (300ng). 24 hours post-transfection either wildtype (WT) or *CARD8* knockout (KO) THP-1s were overlayed on the transfected HEK293T cells in a 1:1 ratio. THP-1s were primed with Pam3CSK4 overnight prior to coculture. Supernatants were harvested 24 hours post coculture to assay for IL-1β secretion as in (**B**). Dotted line indicates limit of detection (LoD). Datasets represent mean ± SD (n=4 biological replicates). p-Values were determined by two-way ANOVA with Dunnett's test using GraphPad Prism 10. ns = not significant, *p<0.05, **p<0.01, ***p<0.001, ****p<0.0001.

The online version of this article includes the following source data for figure 5:

**Source data 1.** Original western blot images used to generate *Figure 5A*.

**Source data 2.** Original western blot .tif files used to generate *Figure 5A*.

## Discussion

We demonstrate that cell-to-cell transmission of HIV-1 from T cells to myeloid cells in both immortalized and primary cell coculture models of infection yields CARD8-dependent inflammasome activation via incoming HIV[PR]. This inflammasome activation occurs in a largely NLRP3-independent manner. In addition, we identified protease inhibitor-resistant strains of HIV-1 that differentially cleave and activate the CARD8 inflammasome. Thus, HIV[PR] mutants selected for their resistance to different protease

inhibitors also affect their ability to cleave host proteins including the inflammasome-forming sensor CARD8.

## CARD8 is the primary inflammasome-forming sensor of HIV-1 infection

Previously, both the NLRP3 and IFI16 inflammasomes had been implicated as innate sensors of HIV-1 infection and drivers of CD4+ T cell depletion using blood and lymphoid-derived CD4+ T cells, respectively, and cell-to-cell transmission was reported to be crucial for IFI16 sensing of abortive HIV transcripts (*Doitsh et al., 2014*; *Galloway et al., 2015*; *Monroe et al., 2014*; *Zhang et al., 2021*). However, the mechanism of NLRP3 inflammasome activation in response to HIV-1 remains elusive. Similarly, there have been reports that IFI16 is not an inflammasome-forming sensor, and instead a nuclear transcriptional regulator of antiviral genes including type I interferons and RIG-I (*Hornung et al., 2009*; *Jiang et al., 2021*; *Thompson et al., 2014*), suggesting that there may be other mechanisms of CD4+ T cell depletion and HIV-dependent inflammasome activation at play. Indeed, CARD8, which is expressed and functional in naïve and memory CD4+ and CD8+ T cells (*Linder et al., 2020*), was recently shown to be required for pyroptosis in primary human blood- and lymphoid-derived CD4+ T cells and humanized mouse models (*Wang et al., 2024*), implicating CARD8 as a major driver of CD4+ T cell depletion during HIV-1 infection. In this study and our prior work (*Kulsuptrakul et al., 2023*), we demonstrate that CARD8 is also the primary innate sensor during HIV-1 infection in myeloid cell types during cell-to-cell transmission. However, our present study does not rule out the possibility that under certain conditions or in certain cell types, NLRP3 inflammasome activation may occur, for example, following GSDMD pore formation following CARD8 inflammasome activation and play a more profound role in promoting HIV-dependent inflammation. Nonetheless, these data, along with other recent work (*Wang et al., 2024*), strongly suggest that CARD8 is a major innate sensor of HIV-1 infection.

## Protease inhibitor resistance mutations and inflammatory disease

Given the important role of HIV^PR in replication, early combination antiretroviral therapy for people living with HIV (PLWH) included protease inhibitors along with RT inhibitors. However, resistance mutations to protease inhibitors arose in PLWH through mutations around the HIV^PR active site allowing for polyprotein processing and viral maturation while avoiding drug inhibition. Despite typically having poor overall viral fitness due to less efficient polyprotein processing and replication relative to wildtype HIV-1 in the absence of protease inhibitors, these mutant drug-resistant HIV-1 strains can persist in PLWH on antiviral therapy (*Luca, 2006*; *Martinez-Picado et al., 1999*; *Prado et al., 2002*; *Resch et al., 2002*). To compensate for mutations in HIV^PR that change its substrate specificity, HIV^gag sometimes evolves mutations around HIV^PR cleavage sites to permit proper polyprotein processing (*Varghese et al., 2013*). Here, we identified multiple HIV^PR inhibitor-resistant strains of HIV-1 that can differentially cleave and activate the CARD8 inflammasome (*Figure 5*, *Supplementary file 1*). As the degree of inflammation is a better predictor of disease progression in untreated individuals than viral load (*Deeks et al., 2004*; *Giorgi et al., 1999*), we speculate that differential CARD8 inflammasome activation could influence disease progression for PLWH harboring HIV^PR resistance mutations that cleave CARD8 more or less efficiently. More broadly, we suggest that host targets of viral proteases like CARD8 may influence the selection of viral variants during treatment with antiviral protease inhibitor monotherapies.

## Viral protease influx activates the CARD8 inflammasome

In this study, we found that HIV-dependent CARD8 inflammasome activation during cell-free infection requires a cationic polymer like DEAE-dextran to facilitate efficient viral infection (see Appendix 1). Despite being infected with the same amount of virus and exhibiting similar percent infection 24 hours post-infection, as measured by intracellular p24^gag, with and without DEAE-dextran, we hypothesize that DEAE-dextran during cell-free infection may increase the total viral dose that enters cells. This increased viral dose in the presence of DEAE-dextran could consist of both infectious particles and non-infectious particles that may nonetheless contain active protease, leading to more efficient viral protease influx to trigger CARD8 sensing. Hence, the percentage of p24^gag-positive cells after 24 hours may be an underestimate of the total amount of viral entry in the DEAE-dextran condition.

We speculate that a considerable influx of incoming HIV$^{PR}$ may be necessary to induce CARD8 inflammasome activation.

We also observed that DEAE-dextran can trigger inflammasome activation in some THP-1 cell lines (see Appendix 1), prompting us to assay for inflammasome activation upon HIV-1 cell-to-cell transmission with target cells at a 1:1 ratio in the absence of cationic polymer. We found that inflammasome activation following cell-to-cell transmission of HIV-1 could be detected by 24 hours (*Figure 3B*), which is delayed relative to our detection of CARD8 inflammasome activation 2 hours post cell-free HIV-1 infection in the presence of DEAE-dextran (*Kulsuptrakul et al., 2023*). Nonetheless, we still hypothesize that this inflammasome activation is driven by active incoming viral protease because treatment with an RT inhibitor has no effect on inflammasome activation, which implies that *de novo* protease production is not necessary (*Figures 3A, B and 4A, B*). Similarly, we observed that IL-1β levels do not increase after plateauing 24 hours after establishing the coculture (*Figures 3B and 4B*), suggesting that secondary infection does not further amplify inflammasome activation. We infer that this is also likely a product of the efficiency of viral entry and the necessity for multiple virions infecting at the same time to deliver a sufficient amount of active HIV$^{PR}$ for cytosolic CARD8 sensing. We postulate that under certain physiological conditions, cell-to-cell transmission can cause CARD8 inflammasome activation when there is an influx of active incoming HIV$^{PR}$ across the viral synapse. Taken together, we speculate that both cell-free infection facilitated by cationic polymer and cell-to-cell transmission can achieve sufficient levels of active HIV$^{PR}$ influx to activate the CARD8 inflammasome.

Macrophages have been reported to be primarily infected through phagocytosis of infected CD4+ T cells or cell-to-cell transmission (*Dupont and Sattentau, 2020*; *Martínez-Méndez et al., 2017*; *Orenstein, 2000*). We demonstrate that unlike CD4+ T cells, which are rapidly depleted by HIV-1 infection and do not release IL-1β or IL-18 (*Linder et al., 2020*), primary macrophages release proinflammatory cytokines in response to HIV$^{PR}$ during cell-to-cell infection from infected primary CD4+ T cells and T cell lines (*Figures 3 and 4*), thus representing a potential source of sustained IL-1β and subsequent chronic immune activation. In addition to promoting chronic immune activation, HIV-dependent IL-1β release from macrophages may also contribute to HIV-1 pathogenesis by activating nearby CD4+ T cells, rendering them susceptible to becoming infected with HIV-1, and thus indirectly promoting CD4+ T cell depletion. Collectively with our prior work (*Kulsuptrakul et al., 2023*), our findings provide further evidence that CARD8 inflammasome activation is driven by incoming HIV$^{PR}$ under conditions where multiple virions may enter cells, and thus could be a potential driver of HIV-1 pathogenesis by promoting chronic immune activation.

## Materials and methods
### Plasmids and reagents
pMD2.G used for HIV-1$_{LAI-VSVG}$ production was a gift from Didier Trono (Addgene). HIV-1$_{LAI}$ has been previously described (*Peden et al., 1991*). The following reagents were obtained through the NIH HIV Reagent Program, Division of AIDS, NIAID, NIH: lopinavir (LPV), nevirapine (NVP), human immunodeficiency virus 1 (HIV-1) NL4-3 BaL Infectious Molecular Clone (p20-36) (HIV-1$_{NL4.3-BaL}$), ARP-11442, contributed by Dr. Bruce Chesebro (*Chesebro et al., 1992*; *Chesebro et al., 1991*; *Toohey et al., 1995*; *Walter et al., 2005*), and Panel of Multi-Protease Inhibitor Resistant Infectious Molecular Clones, HRP-12740, contributed by Dr. Robert Shafer (*Varghese et al., 2013*). Mutant HIV$^{PR}$ sequences were amplified from clinically derived viral cDNA encoding protease genes with resistance to multiple protease inhibitors then cloned into an NL4.3 backbone with overhangs including the 3' end of gag with the gag cleavage site and the 5' end of RT as previously described (*Varghese et al., 2013*). CARD8 variant constructs were cloned as previously described (*Kulsuptrakul et al., 2023*). VX765 and MCC950 were sourced from Invivogen (cat: inh-vx765i-1 and inh-mcc, respectively).

### Cell culture
SUPT1 (ATCC) and THP-1 cells (JK and ATCC) were cultured in RPMI (Invitrogen) with 10% FBS, 1% penicillin/streptomycin antibiotics, 10 mM HEPES, 0.11 g/L sodium pyruvate, 4.5 g/L D-glucose, and 1% GlutaMAX. JK THP-1 cells were used for all experiments in this manuscript and our previous work unless explicitly stated (see Appendix 1; *Kulsuptrakul et al., 2023*). Primary monocytes were cultured in RPMI (Invitrogen) with 10% FBS, and 1% penicillin/streptomycin antibiotics and differentiated in

the presence of 20 ng/mL GM-CSF (Peprotech cat: 300-03) and 20 ng/mL M-CSF (Peprotech cat: 300-25). Primary CD4+ T cells were cultured in RPMI (Invitrogen) with 10% FBS, 1% penicillin/streptomycin antibiotics, and 100 U/mL IL-2. HEK293T (ATCC) lines were cultured in DMEM (Invitrogen) with 10% FBS and 1% penicillin/streptomycin antibiotics. All lines were routinely tested negative for mycoplasma bacteria (Fred Hutch Specimen Processing & Research Cell Bank). SUPT1 and THP-1 cell lines were authenticated by STR profiling analysis (Fred Hutch Genomics core and TransnetYX, Inc) (see Appendix 1).

## HIV-1$_{LAI}$, HIV-1$_{LAI-VSVG}$, and HIV-1$_{NL4.3-BaL}$ production

293T cells were seeded at 2–3×10$^5$ cells/well in six-well plates the day before transfection using TransIT-LT1 reagent (Mirus Bio LLC) at 3 μL transfection reagent/well as previously described (*OhAinle et al., 2018*). For HIV-1 production, 293Ts were transfected with 1 μg/well HIV$_{LAI}$ or HIV-1$_{NL4.3-BaL}$ proviral DNA or 1 μg/well HIV$_{LAI}$ Δenv DNA and 500 ng/well pMD2.G for HIV-1$_{LAI}$, HIV-1$_{NL4.3-BaL}$, and HIV-1$_{LAI-VSVG}$, respectively. One day post-transfection, media was replaced. Two days post-transfection, viral supernatants were collected and filtered through a 20 μm filter and aliquots were frozen at –80°C. HIV-1$_{LAI}$, HIV-1$_{NL4.3-BaL}$ and HIV-1$_{LAI-VSVG}$ proviruses were previously described (*Bartz and Vodicka, 1997*; *Gummuluru et al., 2003*; *Peden et al., 1991*).

## Cell-free and cell-to-cell coculture HIV-1 infection

Cell-free infections with HIV-1$_{LAI-VSVG}$ were done as previously described (*Kulsuptrakul et al., 2023*). Subsequent cell death was assessed by incubating in media containing propidium iodide dye (10μg/mL) for 5 minutes at room temperature then washed once with PBS before fixing with BD CytoFix/Cytoperm (cat: BDB554714) and staining for intracellular p24$^{gag}$ (Beckman Coulter cat#: 6604665) for flow cytometry. In the HIV-1 cell-to-cell transmission system, SUPT1 expressing CCR5 (SUPT1-CCR5) cells were spinoculated at 1100 g for 30 minutes with either HIV-1$_{LAI}$ or HIV-1$_{NL4.3-BaL}$ in the presence of 10μg/mL DEAE-dextran. SUPT1-CCR5 cells were lentiviral transduced to express CCR5 (*Dingens et al., 2017*). After 24 hours, mock- or HIV-1-infected SUPT1-CCR5 cells were washed three times in PBS such that DEAE-dextran and cell-free virus were removed before starting coculture with THP-1 cells or MDMs. THP-1 cells and MDMs were seeded at 5 × 10$^5$ cells/well and primed with 500ng/mL Pam3CSK4 (Invivogen) for 16–24 hours before coculture. Mock or infected SUPT1 cells were seeded at 5 × 10$^5$ cells/well. Cultured supernatants from coculture were harvested 48 hours after starting the coculture for the IL-1R reporter assay, which was previously described (*Kulsuptrakul et al., 2023*).

## Transwell coculture HIV-1 infection

SUPT1 cells were spinoculated at 1100 × $g$ for 30 minutes with HIV-1$_{LAI}$ in the presence of 10 μg/mL DEAE-dextran. After 24 hours, mock- or HIV-1-infected SUPT1 cells were washed three times in PBS and either mixed in a 24-well with THP-1 cells or placed in a transwell insert above target THP-1 cells at a concentration of 5 × 10$^5$ infected SUPT1 cells and 2.5 × 10$^5$ THP-1 cells per well. THP-1 cells were primed overnight with 500 ng/mL Pam3CSK4 before starting coculture. The transwell insert has a 0.4 μm membrane at the bottom of the well (ThinCert Tissue Culture Inserts, Sterile, Greiner Bio-One cat: 665640), allowing virus to pass out of the transwell but not the infected cell. Reverse transcriptase (RT) activity in viral supernatants was measured using the RT activity assay as previously described (*Roesch et al., 2018*; *Vermeire et al., 2012*). A stock of HIV-1$_{LAI}$ virus was titered multiple times, aliquoted at –80°C and used as the standard curve in all assays.

## Monocyte-derived macrophage isolation, differentiation, and editing

Primary monocytes were isolated via negative selection using the EasySep Human Monocyte Isolation Kit (Easy Sep, 1 × 10$^9$) (Stem Cell Technologies) according to the manufacturer's protocols from PBMCs collected from blood donors. Upon isolation, monocytes were seeded at 1 × 10$^6$ cell/mL and differentiated for 5 days in the presence of media containing 20 ng/mL GM-CSF (Peprotech cat: 300-03) and 20 ng/mL M-CSF (Peprotech cat: 300-25), changing media every other day. For edited MDMs, isolated monocytes were electroporated in cuvettes (100 μL) with 2.5–5 × 10$^6$ cells/nucleofection in the presence of pre-complexed Cas9-RNPs (300 pmol sgRNA: 100 pmol Cas9) in Lonza P2 buffer using pulse code DK-100. RNPs were complexed with sgRNA from the Synthego gene KO kit, which includes three sgRNAs per gene. Thus, each sgRNA was present at a 1:1 ratio with Cas9

(QB3 MacroLab or Synthego SpCas9 2NLS Nuclease). A table of sgRNAs used for *AAVS1*, *CARD8*, or *NLRP3* KO can be found in **Supplementary file 2**. After nucleofection, cells were supplemented with 900 µL of prewarmed media and allowed to recover for 30 minutes at 37°C before counting and seeding at ~1–1.5 × 10^6 cells/mL for differentiation. Media was changed 24 hours post nucleofection then differentiated for 5 more days before characterizing knockout efficiency and conducting coculture experiments.

## CD4+ T cell isolation, infection, and coculture

Primary CD4+ T cells were isolated via positive selection using the EasySep Release Human CD4 Positive selection kit (Stem Cell Technologies cat: 17752) according to the manufacturer's instructions from PBMCs collected from blood donors and seeded at 2.5 × 10^6 cells/mL in the presence of 100 U/mL IL-2. T cells were activated 24 hours post-isolation with CD3/CD28 activation beads (Miltenyi Biotech cat: 130-091-441). Activation beads were removed according to the manufacturer's protocols 24 hours later for infection. For infection, T cells were suspended at 1–1.5 × 10^6 cells/mL in 15 mL conical tubes containing 8 µg/mL polybrene, 100 U/mL IL-2, and HIV-1$_{NL4.3-BaL}$ then spinoculated at 1100 × *g* at 30°C for 90 minutes. Three days post-infection, CD4s were assessed for intracellular p24$^{gag}$ via flow cytometry (~10% infected) then washed thrice with PBS before coculturing with MDMs. CD4:MDM coculture RPMI media was supplemented with 100 U/mL IL-2, 20 ng/mL GM-CSF, 20 ng/mL M-CSF, and 500 ng/mL Pam3CSK4.

## CARD8 cleavage assay

HEK293T cells were seeded at 1–1.5×10^5 cells/well in 24-well plates the day before transfection using TransIT-LT1 reagent at 1.5 µL transfection reagent/well (Mirus Bio LLC). One hundred ng of indicated constructs encoding an N-terminal mCherry-tagged CARD8 were co-transfected into HEK293T cells with empty vector ('–'), HIV$_{LAI}$ or PI-R provirus. To normalize HIV$^{gag}$ expression between HIV-1$_{LAI}$ and the PI-R clones, which are in a different vector backbone, 400 ng of HIV-1$_{LAI}$ and 200 ng of all PI-R clones were transfected. All conditions were normalized with empty vector to contain the same amount of DNA. Cytoplasmic lysates were harvested 24 hours post-transfection and immunoblotted as previously described (**Kulsuptrakul et al., 2023**).

## HEK reconstitution assay

HEK293T cells, which endogenously express CARD8, were seeded at 2.25 × 10^5 cell/well in 24-well plates the day before transfection using TransIT-LT1 reagent at 1.5 uL transfection reagent/well (Mirus Bio LLC). Functional inflammasomes were reconstituted by transfecting in 5 ng human CASP1 and 100 ng human IL-1β. To assess the effects of different viral proteases on inflammasome activation, HIV-1$_{LAI}$ or PI-R clones were co-transfected in with CASP1 and IL-1β. As with the CARD8 cleavage assay, a higher amount of 250 ng HIV-1$_{LAI}$ was added relative to the PI-R clones, which were all added at 125 ng, to normalize HIV$^{gag}$ expression between the different vector backbones. All conditions were normalized with empty vector to contain the same amount of DNA. Cultured supernatant was harvested 24 hours post-transfection to assay for IL-1β secretion via IL-1R reporter assay.

## Acknowledgements

We thank everyone in the Emerman and Mitchell labs for helpful feedback on the project, Terry Hafer and Marisa Yonemitsu for critical reading of the manuscript, Liang Shan and his lab members for discussions and sharing of unpublished results and protocols, and the Fred Hutchinson Shared Resources Genomics, Flow Cytometry, and Specimen Processing & Research Cell Bank cores. LPV (HRP-9481), NVP (HRP-4666), and p24$^{gag}$ antibody (ARP-3537) were provided by the AIDS Reagent Program, Division of AIDS, NIAID, and NIH. PSM is an HHMI Freeman Hrabowski Scholar and is supported by grants from the National Institutes of Health (NIH) (DP2 AI 154432-01) and the Mallinckrodt Foundation to PSM. ME is supported by NIH grant DP1 DA051110. JK is supported by the University of Washington Cellular and Molecular Biology Training Grant (T32 GM007270).

## Additional information

### Funding

| Funder | Grant reference number | Author |
|---|---|---|
| National Institutes of Health | DP2 AI 154432-01 | Patrick S Mitchell |
| National Institutes of Health | DP1 DA051110 | Michael Emerman |
| National Institutes of Health | T32 GM007270 | Jessie Kulsuptrakul |
| Howard Hughes Medical Institute | | Patrick S Mitchell |
| Mallinckrodt Foundation | | Patrick S Mitchell |

The funders had no role in study design, data collection and interpretation, or the decision to submit the work for publication.

### Author contributions

Jessie Kulsuptrakul, Conceptualization, Resources, Data curation, Formal analysis, Validation, Investigation, Visualization, Methodology, Writing – original draft, Writing – review and editing; Michael Emerman, Patrick S Mitchell, Conceptualization, Resources, Data curation, Formal analysis, Supervision, Funding acquisition, Visualization, Writing – original draft, Project administration, Writing – review and editing

### Author ORCIDs

Jessie Kulsuptrakul https://orcid.org/0000-0003-3881-4686
Michael Emerman https://orcid.org/0000-0002-4181-6335
Patrick S Mitchell https://orcid.org/0000-0001-8375-9060

Joint Public Review: https://doi.org/10.7554/eLife.102676.3.sa1
Author response https://doi.org/10.7554/eLife.102676.3.sa2

## Additional files

### Supplementary files

Supplementary file 1. Protease inhibitor-resistant (PI-R) clones assayed in *Figure 5* with corresponding mutations in HIV protease (HIV[PR]) and HIV[gag]. [1]These clones were previously cloned and assayed for PI-R in *Varghese et al., 2013*. The PI-R subset used in *Figure 5B* are bolded and highlighted in red or green and denote either hypo- or hyper-active CARD8 cleavage, respectively. The last column reports additional amino acid changes in the PI-R clones that were observed via whole plasmid Oxford Nanopore sequencing. *We were unable to sequence verify PI-R3 due to poor plasmid quality. NFV, nelfinavir; FPV, fosamprenavir; SQV. saquinavir; IDV, indinavir; LPV, lopinavir; TPV, tipranavir; DRV, darunavir. The consensus subtype B sequence can be found on the Stanford HIV Drug Resistance Database (HIVDB) (*Stanford University HIV Drug Resistance Database, 2025*). Relative CARD8 cleavage was determined by quantifying band volume of the CARD8 cleavage product in BioRad Image Lab 6 and comparing to cleavage with HIV-1$_{LAI}$.

Supplementary file 2. sgRNAs used in this study.

MDAR checklist

### Data availability

All data generated during this study are included in the manuscript or posted to Dryad. Sequences have been submitted to GenBank at the following accession codes (PV761644–PV761650).

The following datasets were generated:

| Author(s) | Year | Dataset title | Dataset URL | Database and Identifier |
|---|---|---|---|---|
| Kulsuptrakul J, Emerman M, Mitchell PS | 2025 | Data from: CARD8 inflammasome activation during HIV-1 cell-to-cell transmission | https://doi.org/10.5061/dryad.qbzkh18vn | Dryad Digital Repository, 10.5061/dryad.qbzkh18vn |
| Kulsuptrakul J, Emerman M | 2025 | Mutant HIV-1 clone CA126802 from USA gag protein (gag) and pol protein, protease region, (pol) genes, partial cds | https://www.ncbi.nlm.nih.gov/nuccore/PV761644 | NCBI GenBank, PV761644 |
| Kulsuptrakul J, Emerman M | 2025 | Mutant HIV-1 clone CA122805 from USA gag protein (gag) and pol protein, protease region, (pol) genes, partial cds | https://www.ncbi.nlm.nih.gov/nuccore/PV761645 | NCBI GenBank, PV761645 |
| Kulsuptrakul J, Emerman M | 2025 | Mutant HIV-1 clone CA96457 from USA gag protein (gag) and pol protein, protease region, (pol) genes, partial cds | https://www.ncbi.nlm.nih.gov/nuccore/PV761646 | NCBI GenBank, PV761646 |
| Kulsuptrakul J, Emerman M | 2025 | Mutant HIV-1 clone CA96458 from USA gag protein (gag) and pol protein, protease region, (pol) genes, partial cds | https://www.ncbi.nlm.nih.gov/nuccore/PV761647 | NCBI GenBank, PV761647 |
| Kulsuptrakul J, Emerman M | 2025 | Mutant HIV-1 clone CA50834-1 from USA gag protein (gag) and pol protein, protease region, (pol) genes, partial cds | https://www.ncbi.nlm.nih.gov/nuccore/PV761648 | NCBI GenBank, PV761648 |
| Kulsuptrakul J, Emerman M | 2025 | Mutant HIV-1 clone CA96451 from USA gag protein (gag) and pol protein, protease region, (pol) genes, partial cds | https://www.ncbi.nlm.nih.gov/nuccore/PV761649 | NCBI GenBank, PV761649 |
| Kulsuptrakul J, Emerman M | 2025 | Mutant HIV-1 clone CA20392-1 from USA gag protein (gag) and pol protein, protease region, (pol) genes, partial cds | https://www.ncbi.nlm.nih.gov/nuccore/PV761650 | NCBI GenBank, PV761650 |

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

## Appendix 1

There have been four different sublines of THP-1 cells previously characterized (*Kasai et al., 2022*). Using short tandem repeat (STR) profiling, we were able to distinguish the WT THP-1 cell stocks that were used as the parental line for knockouts and complemented knockouts in this work and our prior work (*Kulsuptrakul et al., 2023*), as distinct from WT THP-1 cells sourced from ATCC at three different loci (*Appendix 1—figure 1A*). Of note, unlike the THP-1 cells used here (referred to as JK THP-1), ATCC THP-1 cells elicited IL-1β secretion in the absence of HIV-1 infection in the presence of 20μg/mL DEAE-dextran (*Appendix 1—figure 1B*). Unless otherwise specified, any mention of 'THP-1 cells' are referring to our JK THP-1 cells, not ATCC THP-1 cells. Nonetheless, given the sensitivity of some THP-1 sublines to elicit an inflammasome response in the presence of DEAE-dextran, we assessed whether or not we could establish systems to measure HIV-1-induced CARD8-dependent inflammasome activation in the absence of DEAE-dextran. Thus, we infected either wildtype (WT) or *CARD8* knockout (KO) THP-1 cells with wildtype HIV-1$_{LAI}$ in either the presence or absence of DEAE-dextran and measured cell death and IL-1β secretion 24 hours post-infection as readouts of inflammasome activation. We found that despite achieving similar levels of infection (20–30%) as measured by intracellular p24$^{gag}$ after spinoculation with and without DEAE-dextran (*Appendix 1—figure 1C*, left), we detected robust CARD8-dependent inflammasome activation in WT THP-1 cells infected only in the presence of DEAE-dextran (*Appendix 1—figure 1C*, middle and right). These data suggest that cationic polymer is necessary to observe HIV-dependent CARD8 inflammasome activation in our cell-free system.

**A**

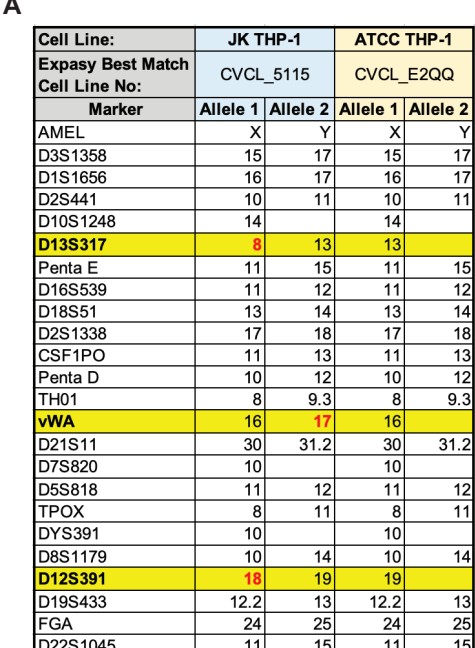

**B**

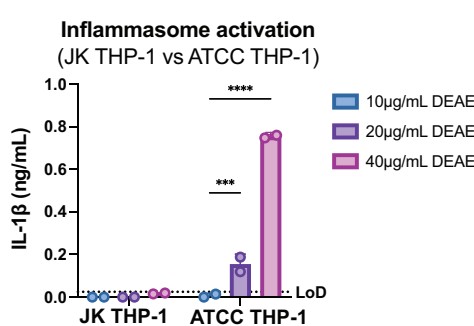

**C**

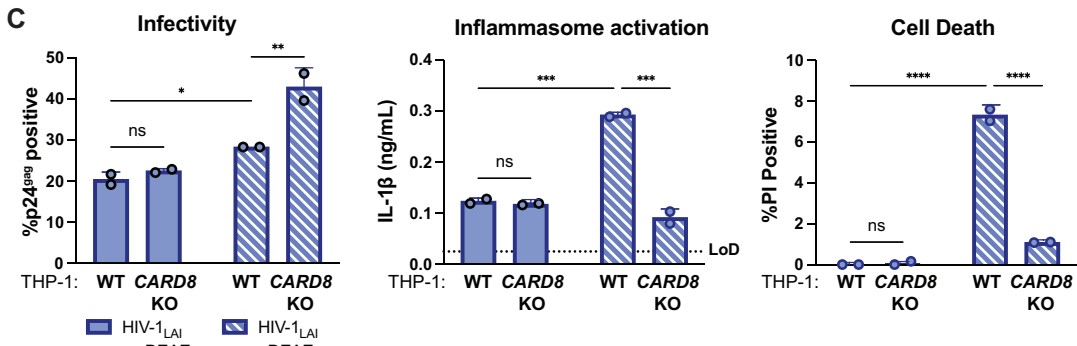

**Appendix 1—figure 1.** Characterization of THP-1 cells. (**A**) Promega GenePrint 24 system STR analysis summary of our JK THP-1 cells versus ATCC THP-1 cells. Cell line authentication was done by TransnetYX, Inc by following the protocol described in ANSI/ATCC ASN-0002–2011. The STR alleles were searched on the ATCC Database and the Expasy best match cell numbers for each cell line had a 100% database match. Distinguishing loci are highlighted in yellow and distinguishing alleles are in red. (**B**) JK and ATCC THP-1 cells were primed with Pam3CSK4 overnight then treated with increasing doses of DEAE-dextran for 24 hours before probing for IL-1β secretion. (**C**) Wildtype (WT) or *CARD8* knockout (KO) THP-1 cells were infected with wildtype HIV-1$_{LAI}$ at the same MOI in the presence or absence of DEAE-dextran (10 µg/mL) then harvested after 24 hours and assayed for: left, percent infection via intracellular p24$^{gag}$; middle, inflammasome activation by IL-1β secretion via IL-1R reporter assay; and right, cell death via propidium iodide (PI) dye uptake using flow cytometry. %PI positive and IL-1 levels are normalized to mock control. Dotted line indicates limit of detection (LoD). Datasets represent mean ± SD (n=2 biological replicates). Two-way ANOVA with (**B**) Sidak's or (**C**) Tukey's test using GraphPad Prism 10. ns = not significant, *p<0.05, **p<0.01, ***p<0.001, ****P<0.0001.

