## [Editor Report · eLife Assessment]

In this study, the authors investigated whether HIV-1 cell-to-cell transmission activates the CARD8 inflammasome in macrophages. The data **convincingly** support the idea that CARD8 is activated by the viral protease, promoting inflammation. The study's significance is further enhanced by including time-course analyses in primary T cells and macrophages and provides **valuable** insights into the role of CARD8 in HIV-induced inflammation.

---

## [Referee Report · Joint Public Review]

Following up on their previous work, the authors investigated whether HIV-1 cell-to-cell transmission activates the CARD8 inflammasome in macrophages, a key question given that inflammasome activation in myeloid cells triggers proinflammatory cytokine release. Co-cultures of HIV-infected T cells with macrophages led to viral spreading, resulting in IL1β release and cell death, with CARD8 playing a crucial role in this inflammasome response, triggered by HIV protease. The authors also found that HIV isolates resistant to protease inhibitors showed differences in CARD8 activation and IL1β production, highlighting the clinical relevance of their findings. Overall, this well-written study provides strong evidence for the role of CARD8 in protease-dependent sensing of viral spread, with implications for understanding chronic inflammation in HIV infections and its potential contribution to systemic immune activation, especially under ART. The authors have addressed initial weaknesses and verified effects in cocultures of primary T cells and macrophages. They now also provide evidence that CARD8 is activated by protease from incoming viral particles. Further studies are needed to clarify how much this mechanism contributes to systemic immune activation in untreated infections and whether this mechanism drives chronic inflammation under ART.

---

## [Author Response]

The following is the authors’ response to the current reviews.

We again thank you for the positive and constructive feedback on our manuscript, and for highlighting its contributions to understanding the role of CARD8 in viral protease-triggered sensing of viral spread, and the potential impact of our findings on chronic inflammation and immune activation. We agree that it will be important for future work to address whether or not HIV-1 protease-triggered CARD8 inflammasome activation contributes to chronic inflammation in PLWH who are receiving ART.

In response to the question about the baseline level of IL-1β in Fig. 4D, the figure below shows the mock condition for the CD4+ T cell:MDM coculture. We had done this control in parallel with the data presented in the submitted figure. Levels of IL-1β during HIV-1 infection are increased over background (i.e., mock infection). We note that for donor G the IL-1β concentration is below the limit of detection for this assay. Thus, it remains possible that other inflammasomes contribute modestly during cell-to-cell transmission of HIV-1; however, incomplete knockout of *CARD8* in a minority of cells may also contribute to the observed levels of IL-1β in response to HIV-1 infection. Nonetheless, collectively, our data strongly supports the role for CARD8 in HIV-1 protease-triggered inflammasome activation.

The following is the authors’ response to the original reviews.

**Joint Public Review:**
Following up on their previous work, the authors investigated whether cell-to-cell transmission of HIV-1 activates the CARD8 inflammasome in macrophages, an important question given that inflammasome activation in myeloid cells triggers proinflammatory cytokine release. The data support the idea that CARD8 is activated by the viral protease and promotes inflammation. However, time-course analyses in primary T cells and macrophages and further information on the specific inflammasome involved would further increase the significance of the study.Strengths:The manuscript is well-written and the data is of good quality. The evidence that CARD8 senses the HIV-1 protease in the context of cell-to-cell transmission is important since cell-to-cell transmission is thought to play a key role in viral spread in vivo, and inflammation is a major driver of disease progression. Clean knockout experiments in primary macrophages are a notable strength and the results clearly support the role of CARD8 in protease-dependent sensing of viral spread and the induction of IL1β release and cell death. The finding that HIV-1 strains are resistant to protease inhibitors differ in CARD8 activation and IL1β production is interesting and underscores the potential clinical relevance of these results.Weaknesses:One weakness is that the authors used T cell lines which might not faithfully reflect the efficiency of HIV-1 production and cell-cell transfer by primary T cells. To assess whether CARD8 is also activated by protease from incoming viral particles earlier time points should be analyzed. Finally, while the authors exclude the role of NLRP3 in IL-1b and the death of macrophages it would be interesting to know whether the effect is still Gasdermin D dependent.
**Recommendations for the authors**
(1) Co-culture assay should also be done between primary CD4 cells and primary MDMs, because T-cell lines produce much more viruses, and the efficiency of cell-tocell transmission might be dramatically different in primary cells compared to cell lines.

We have now added data from experiments using infected primary CD4 cells as the donor cells in cell-to-cell HIV-1 transmission to MDMs in new Figure 4. The results largely phenocopy the SUPT1:MDM coculture in that we observe inflammasome activation after co-culture of HIV-infected primary T cells with primary MDMs. We find that this inflammasome activity induced by the CD4:MDM cell-to-cell transmission is abrogated by knockout of CARD8 in the MDMs or treatment of HIV protease inhibitor lopinavir (LPV) or caspase 1 inhibitor VX765, suggesting that this activation is dependent on CARD8, HIV protease, and caspase 1. Additionally, the signal persists in the presence of reverse transcriptase inhibitor nevirapine (NVP), suggesting that the incoming protease is driving activation.

(2) For all co-culture experiments, supernatants were collected at 48 or 72 hours. Since CARD8 activation is expected to be driven by incoming viral particles without RT, they should measure cytokine production at much earlier time points. 2-3 days co-culture raises concerns. Ideally, the authors can provide a time-course.

We have now added a time course of the SUPT1:MDM coculture from 3 unique donors taken at 4, 24, 48, and 72 hours post coculture in the presence or absence of reverse transcriptase inhibitor (see new Figure 3) as well as for the primary CD4 cells to MDM co-culture (see new Figure 4). We detect IL-1β at the 24hour time point (and later), but not at the 4-hour time point which is slower than what was detected by direct cell-free infection (Kulsuptrakul et al., 2023). However, we still hypothesize that this is driven by active incoming viral protease because the signal is not abrogated by a reverse transcriptase inhibitor, which indicates that *de novo* protease production is not necessary. We also observed that IL-1β levels do not increase after plateauing 24h after establishing the co-culture, suggesting that secondary infection does not further amplify inflammasome activation. We now speculate on this in the Discussion.

(3) A potential confounder in the data in Figure 4 is that despite rightly including the cognate adaptations in the Gag cleavage sites with the PI-R protease mutants, some of these viruses still display Gag processing defects. Can the authors disentangle the potency of PR mutant cleavage with either reduced cell entry or reduced protease availability due to processing defects in the incoming virions?

The reviewer is correct that although the western blot with the p24^gag^ antibody suggests that Gag is processed, we cannot rule out that other variables do not contribute to the observed difference in CARD8 inflammasome activation. For example, PI-R clones relative to the LAI strain may have distinct protease substrate specificity, variable efficiency/kinetics in viral assembly, gag dimerization, and other factors may ultimately influence CARD8 inflammasome activation. We have updated the text to reflect these possibilities. Nonetheless, this argument does not change the conclusion that CARD8 inflammasome activation is affected by protease mutations acquired during drug resistance.

(4) There is considerable donor variation in the macrophages (unsurprising) but can the authors correlate this with CARD8 expression and are there any off-target effects on macrophage permissivity to HIV-1 infection?

We have now considerably increased the number of primary cell donors from the first submission (see Author response table 1 below). We find that the non-responsive donor presented in the first submission is aberrant since all others do respond to a greater or lesser degree (Figure 3, Figure 4). However, the reviewer may be correct that the particular aberrant donor MDMs were poorly infected. We also note that despite donor variability in the degree of activation (IL-1β secretion) from cocultures with HIV_BaL_-infected SUPT1 cells, HIV-induced activation is comparable to the activation induced by VbP (see new Figure 3—figure supplement 1). We do not see a notable difference in CARD8 expression between donors. Nonetheless, with the added number of primary cell donors, the data are consistent with a role of primary MDMs from nearly all donors in supporting a CARD8-dependent, HIV-protease dependent inflammasome response after co-culture with infected T cells. We have left in data from all of the donors so that readers can appreciate the variability among primary cells.

**Author response table 1. sa2table1:** 

	Preprint	Resubmission
# of WT MDM donors	3 (Figure 3A)	9 (Figure 3A and B, Figure 4A and B)
# of KO MDM donors	3 (Figure 3C and D: donor 7-9)	7 (Figure 3C and D, Figure 4C and D donor 7-12,15)
# of SUPT1:MDM donors	6 (Figure 3)	11 (Figure 3)
# of CD4:MDM donors	0	4 (Figure 4)

In addition, to address the reviewer concerns about off-target effects of the sgRNAs on macrophage permissivity, we assessed our CD4:MDM cocultures for percent infectivity via intracellular p24^gag^ in *AAVS1* vs *CARD8* KO MDMs and we observed no significant difference in infectivity in *AAVS1* vs *CARD8* KO MDMs see Author response image 1 of MDMs after co-culture with T cells that is not affected any potential off-target effects of the sgRNAs.

**Author response image 1. sa2fig1:** Equivalent infection in AAVS1 vs CARD8 KOMDMs.

(5) The authors suggest that NLRP3 is unlikely to be the mediator of IL-1b and cell death in the macrophages. Is this death still GSDMDdependent, what other NLRs are expressed in this system and does it make a difference what PAMP you use to prime the response?

We have now added additional data in support of the conclusion that NLRP3 is not a mediator of the IL-1β secretion in the infected SUPT1 cells to primary MDMs coculture. In addition to using an NLRP3 inhibitor, we have now also made *NLRP3* KOs MDMs and used these in the coculture experiments which show that the IL-1β secretion after coculture of infected SUPT1 cells and primary MDMs is mediated by CARD8 and not NLRP3 because the signal is abrogated by CARD8 knockout, but not by NLRP3 knockout. This new data is shown in Figure 3.

To assess the role of GSDMD, we treated SUPT1:MDM cocultures with disulfiram, a GSDMD inhibitor (Hu et al., 2020). Disulfiram treatment abrogated IL-1β secretion, suggesting that this activation is indeed GSDMD-mediated (see Author response image 2 below). We choose not to include the disulfiram result in the final manuscript since we have not ruled out cytotoxic effects of the drug.

There are likely other NLRs expressed in primary MDMs; however, since inflammasome activation is completely absent in the *CARD8* KO MDMs, we infer that CARD8 is the main inflammasome-forming sensor in this system. However, we cannot rule out the possibility of other innate sensors being activated downstream of CARD8 or under different differentiation conditions.

To address the concern that alternative priming affects CARD8 activation, we compared pre-treatment of cells with Pam3CSK4 or lipopolysaccharide (LPS) in the presence or absence of HIV protease inhibitor and reverse transcriptase inhibitor. Regardless of the priming agent used, we observed HIV protease-dependent activation that persisted in the presence of reverse transcriptase inhibitor, suggesting that CARD8 is the main sensor under LPS and Pam3CSK4 priming (new Figure 3—figure supplement 1).

**Author response image 2. sa2fig2:** Inflammasome activation following cell-to-cell HIV infection is mediated by GSDMD. SUPT1-CCR5 cells were either mock-infected or infected with HIV-1_NL4.3BaL_ for 20 hours before coculturing with MDMs in either the presence or absence of GSDMD inhibitor disulfarim (25μM). Cocultures were harvested 24 hours later to assess (left) IL-1β secretion via IL-1 reporter assay and (right) cell viability via CellTiter-Glo assay. Viability was calculated by normalizing to relative luminescence units in the mock untreated control. Dotted line indicates limit of detection (LoD). Dashed line indicates 100% viability as determined by untreated mock control. Datasets represent mean ± SD (n=2 technical replicates for one donor). Two-way ANOVA with Sidak’s test using GraphPad Prism 10. ns = not significant, *p<0.05,**p<0.01, ***p<0.001, ****p<0.0001.

Minor points(1) In Figure 1, the authors should clarify whether LAI or LAI-VSV-G was used.

Wild-type virus (LAI strain) was used in Figure 1. This has now been clarified in the figure legend.

(2) In Figure 1, the fraction of infected cells without DEAE was ~20% in both WT and CARD8 KO THP-1, suggesting somewhat efficient viral entry even in the absence of DEAE. How do the authors reconcile this with the lack of IL-1β production? The increase in infection observed in WT THP-1 +DEAE was overall modest (from ~20% to 25-30%) compared to the dramatic difference in IL-1β production. Can they provide more evidence or discuss how DEAE might be impacting cytokine production? If differences in viral entry are the explanation for differences in inflammasome activation, then they should be able to overcome this by using virus at a higher MOI in the absence of DEAE. Experiments proposed in Figure 1 +/- DEAE should be repeated using a range of MOI for LAI and showing the corresponding percent infection in THP-1 cells (which is not shown in Figure S2 for LAI-VSVG).

We hypothesize that the lack of IL-1β production without DEAE is likely due to an insufficient amount of incoming viral protease to induce CARD8 activation. Though the increase in infection with DEAE is modest by intracellular p24^gag^ at 24 hours post infection, we infer that intracellular p24^gag^ may be largely underestimating the actual increase in viral efficiency achieved with DEAE (now in Supplemental Note). We have also updated Figure S2 (now Figure 2—figure supplement 1) legend to include the percent infection for HIV-1_LAI_ and HIV-1_LAI-VSVG_ infections. We agree that activation in the absence of DEAE could be overcome by infecting with a more concentrated viral stock to increase the MOI. Indeed, our decision to use the cell-to-cell transmission model achieves this in a more physiologic context.

(3) In Figure S1, the authors point out that RT-activity in the supernatants was similar in the cell-free vs. cell-to-cell model. While in the transwell system THP-1 cells are the only cells capable of producing new virions, how are they able to differentiate viral production from sup-T1 vs. THP-1 in the cell-to-cell system? At a minimum, they should provide some data on the observed RT activity in matching wells containing the same number of infected sup-T1 cells utilized in coculture experiments.

We think this may have been a misinterpretation. In Figure S1 (now Figure 1), we compare the amount of virus available in the lower chamber of the transwell versus the cell-to-cell condition. We are not comparing cell-free to cell-to-cell infection. We have changed the text and figure title to clarify this point.

(4) Can the authors provide additional comments on the lack of IL-1β release in donor C in Figure 3? The donor did not produce IL-1β in response to VbP or HIV, although the WB for CARD8 appears similar to the other two donors.

We have now tested MDMs from additional donors and continue to find a range of IL-1β secretion after the coculture. However, donor C is aberrant since each of the other donors had detectable IL-1β secretion in response to VbP and HIV-1 to greater or lesser extents. Nonetheless, we have included additional donors summarized in the table above corresponding to major comment #4.

(5) For Figure 3, can the authors provide information on the fraction of MDMs that were infected after coculture with sup-T1 cells? Why didn't the authors measure cell death in MDMs?

It is difficult to measure the fraction of MDMs infected or dying in the cocultures since it is hard to separate signal from the T cells. Although it would be possible to do so, in this manuscript, we instead prefer to focus on the potential contribution of CARD8 inflammasome activation in exacerbating chronic inflammation in response to HIV rather than the depletion of macrophages.

(6) In Figure 4, did the authors introduce the mutations associated with PI resistance into the same LAI backbone? If not, this is not a fair comparison, as viral protein expression levels were not at the same level, indicated in Figure 4A. Additionally, such comparison will be further strengthened by using cells other than 293T cells for the coculture assay.

No, we did not introduce these mutations into LAI, since they were already in an NL4.3 backbone and NL4.3 and LAI differ by only 1 amino acid in protease. We have updated Table S1 to report this amino acid difference. We also note that in our previous manuscript we tested much more diverse proteases such as a clade A HIV-1, HIV-2, and SIVs and find comparable CARD8 cleavage to LAI.

Additions not requested by Reviewers:

THP-1 characterization

In our previous work, we noticed that different “wildtype” THP-1 lines behaved uniquely in response to DEAE-dextran. In particular, we observed inflammasome activation in response to DEAE-dextran alone at the concentration used for spinoculations (20μg/mL), whereas the other THP-1 line did not. Thus, we performed STR profiling on each THP-1 cell line and determined that the THP-1 cells used in our studies (JK THP1s) are distinct from THP-1 cells from ATCC at 3 different loci. This data is now included in the Supplemental Note (Appendix 1—figure 1). Please note that all data in this and the accompanying manuscript were performed in JK THP-1 cells.

Whole plasmid sequencing of the PI-resistant HIV clones

Since preprint submission, we have done whole plasmid Oxford Nanopore sequencing on the PI-resistant HIV clones obtained from the NIAID HIV/AIDS Specimen Repository Program. Of note, there were a handful of previously unreported mutations included in these plasmid stocks within protease. We have updated Table S1 to include an additional column titled “Additional amino acid changes in HIV^PR^ relative to NL4.3.”

References

Hu JJ, Liu X, Xia S, Zhang Z, Zhang Y, Zhao J, Ruan J, Luo X, Lou X, Bai Y, Wang J, Hollingsworth LR, Magupalli VG, Zhao L, Luo HR, Kim J, Lieberman J, Wu H. 2020. FDA-approved disulfiram inhibits pyroptosis by blocking gasdermin D pore formation. Nat Immunol 21:736–745. doi:10.1038/s41590-020-0669-6

Kulsuptrakul J, Turcotte EA, Emerman M, Mitchell PS. 2023. A human-specific motif facilitates CARD8 inflammasome activation after HIV-1 infection. eLife 12:e84108. doi:10.7554/eLife.84108